# Ginseng Stem-Leaf Saponins in Combination with Selenium Promote the Immune Response in Neonatal Mice with Maternal Antibody

**DOI:** 10.3390/vaccines8040755

**Published:** 2020-12-11

**Authors:** Yong Wang, Lijia Yuan, Xuemei Cui, Wei Xu, Sijia Fang, Zoushuyi Li, Meiqian Lu, Ye Wu, Xiaodan Ma, Xiaoqing Chi, Songhua Hu

**Affiliations:** Department of Veterinary Medicine, College of Animal Sciences, Zhejiang University, 866 Yu Hang Tang Rd., Hangzhou 310058, China; wyuan526@zju.edu.cn (Y.W.); 11617034@zju.edu.cn (L.Y.); maycui@zju.edu.cn (X.C.); zhihong0902@zju.edu.cn (W.X.); fangsj@zju.edu.cn (S.F.); 21817088@zju.edu.cn (Z.L.); 21717084@zju.edu.cn (M.L.); 3130100283@zju.edu.cn (Y.W.); maxiaodan@zju.edu.cn (X.M.); tiffany1127@zju.edu.cn (X.C.)

**Keywords:** ginseng stem-leaf saponins, selenium, attenuated pseudorabies virus vaccine, adjuvant, neonatal mice, maternal antibody

## Abstract

Neonates acquire from their mothers maternal antibody (MatAb) which results in poor immune response to vaccination. We previously demonstrated that ginseng stem-leaf saponins in combination with selenium (GSe) had adjuvant effect on the immune response to an attenuated pseudorabies virus (aPrV) vaccine. The present study was to evaluate GSe for its effect on the immune response to aPrV vaccine in neonatal mice with MatAb. Results showed that GSe had adjuvant effect on the immune response to aPrV vaccine in neonates. When GSe was co-administered with aPrV vaccine (aP-GSe), specific gB antibody, Th1 cytokines (IL-2, IL-12 and IFN-γ) and Th2 cytokines (IL-4, IL-6 and IL-10) responses were significantly increased in association with enhanced protection of vaccinated neonates against the lethal PrV challenge even though MatAb existed when compared to the neonates immunized with aPrV vaccine alone. GSe-enhanced immune response depended on its use in the primary immunization. The mechanisms underlying the adjuvant effect of GSe may be due to more innate immune related pathways activated by GSe. Transcriptome analysis of splenocytes from neonates immunized with aP-GSe, aPrV or saline solution showed that there were 3976 differentially expressed genes (DEGs) in aP-GSe group while 5959 DEGs in aPrV group when compared to the control. Gene ontology (GO) terms and Kyoto encyclopedia of genes and genomes (KEGG) pathways analysis showed that innate immune responses and cytokine productions related terms or pathways were predominantly enriched in aP-GSe group, such as “NOD-like receptor signaling pathway”, “Natural killer cell mediated cytotoxicity”, “NF-κB signaling pathway”, “cytokine-cytokine receptor interaction”, and “Th1 and Th2 cell differentiation”. Considering the potent adjuvant effect of GSe on aPrV vaccine in neonatal mice with MatAb, it deserves further investigation in piglets.

## 1. Introduction

Neonates and infants are immunologically naive and more susceptible than adults to infectious diseases [1,2,3]. Enhancement of the passive transfer of maternal antibodies (MatAb) by immunization of the mother is an important strategy to prevent neonates from infection at the early stage of life [4]. However, MatAb is a double-edged sword in young animals: MatAb provides protection while it also interferes with immune responses to vaccination [1,5,6].

Pseudorabies (Pr) is an infectious disease caused by porcine herpesvirus-1 (SuHV-1), pseudorabies virus (PrV), or Aujeszky’s disease virus. PrV usually causes pigs infected, leading to huge amount of economic losses in the pig industry [7]. The infection of PrV is fatal in piglets and the mortality approaches 100%. Typical manifestation is paralyzed hind limbs which makes the piglets sit in a “dog-like” position, walk in circles or lay recumbent and paddle. The infected piglets often die from disorder of the central nervous system within 24 to 36 h [8,9]. Immunization of sows and piglets to protect neonates from infection is a common approach to PrV control [10,11]. However, due to the immature immune system of neonates and the immune suppressive effect of MatAb, immunization of neonates often result in reduced, modified or undetectable immune responses in various mammalian species including pigs [12]. After analysis of 14801 swine serum samples in 14 provinces of China during 2012 to 2013, Zou et al. found that 54.29% of pig herds (158/291) were infected with PrV although the animals had received PrV vaccination [13].

There is a period when neonates are susceptible to infectious diseases after MatAb declines and before vaccine-induced immune response arise [1,14]. Pomorska-Mol et al. observed that serum PrV specific antibody was below the protection level for about 2 weeks when piglets with MatAb were immunized with PrV vaccine at 12 weeks old (*n* = 14); however, when piglets with MatAb were inoculated with PrV vaccine for one immunization (8 weeks old; *n* = 13) or twice immunizations (1 and 8 weeks old; *n* = 13), the PrV specific antibody response was below the protection level with a period of at least 8 weeks [15,16]. Therefore, searching for an approach to shorten or close the window of susceptibility is of significance for the control of animal infectious diseases.

We previously demonstrated that ginseng stem-leaf saponins in combination with selenium (GSe) could significantly improve the immune responses against an attenuated PrV (aPrV) vaccine in a mouse model [17]. In the present study, we evaluated the adjuvant effect of GSe on the immune responses to aPrV vaccine and searched the possibility for shortening or closing the window of susceptibility to infection diseases using GSe in neonatal mice with MatAb by measuring the antibody response and cytokine production. RNA-seq technology was also used to explore the mechanisms underlying of GSe as an adjuvant.

## 2. Materials and Methods

### 2.1. Animals

Six to eight weeks old female and male ICR mice were purchased from Shanghai Laboratory Animal Center Co., Ltd. (Shanghai, China). Mice were housed in cages in an environment with controlled temperature (24 ± 1 °C) and humidity (50 ± 10%) and with ad libitum food and water.

### 2.2. Ethical Statement

All the experiments pertaining to animals use and their care strictly followed the Guidelines of Laboratory Animals of Zhejiang University and all the protocols were approved by Zhejiang University Animals Ethics Committee (ZJU20160377).

### 2.3. Vaccines and Adjuvant

Inactivated pseudorabies virus (InPrV) vaccine were purchased from Wuhan Keqian Biology Co., Ltd. (Wuhan, China). Attenuated pseudorabies (aPrV) vaccine was the product of Boehringer Ingelheim (Vetmedica, Inc., Duluth, GA, USA). Standardized ginseng stem-leaf saponins (GSLS) according to Chinese Veterinary Pharmacopoeia were purchased from Hongjiu Ginseng Industry Co., Ltd. (Jilin, China). According to HPLC analysis, it contained ginsenosides Re (16.36%), Rd (9.0%), Rg1 (6.0%), Rb2 (3.8%), Rc (3.7%), Rb1 (2.4%), Rf (0.1%).

Sodium selenite (medical grade) was purchased from Jinping Chemical Technology Co., Ltd. (Shanghai, China). GSLS (0.3 mg) and Se (0.1 mg) were dissolved in 10 mL sterile saline solution and served as GSe solution. GSe solution was sterilized by passing through a 0.22 µm filter and the endotoxin level of solutions was below 0.5 EU/mL.

### 2.4. Experimental Design for Transfer of Maternal Antibody from Mother to Neonate

As shown in Figure 1A, female mice (*n* = 8) were intramuscularly (i.m.) inoculated with InPrV vaccine twice at 14 days interval. They were mated with male mice at 14 days post the booster vaccination and delivered approximately 21 days later. Female mice (*n* = 8) injected with saline solution served as control (non-immunized). After parturition, half of neonates were nursed by their biological mother and the other half were nursed by non-biological mother [18]. Neonates were weaned on day 21 after birth. Blood samples from mothers and neonates for PrV gB specific antibody analysis were collected at 7, 14, 21, 28 days after birth.

### 2.5. Experimental Design for Effect of GSe on the Immune Responses to aPrV Vaccine in Neonates with MatAb about 50% of Blocking Rate

As shown in Figure 2a, female mice (*n* = 10) were received 0.1 mL of InPrV vaccine (diluted 100 folds) twice at 14 days interval and mated with male mice at 14 days after the booster immunization. A total of 60 neonates (weaned on day 21 after birth) with blocking rate of gB antibody at 45–55% were used and divided into 3 groups (*n* = 20/group). Then neonates were i.m. injected with an aPrV vaccine (1000 TCID_50_, 50 μL) diluted in saline solution with GSLS and Se (aP-GSe) or saline solution (aPrV). Neonates received saline solution only served as control group. Blood from mothers and neonates were sampled every two days to analyze gB antibody levels. Then neonates (*n* = 10/group) were challenged with intraperitoneal (i.p.) injection of fPrV (5 × 10^5^ TCID_50_) on day 28 or 49.

### 2.6. Experimental Design for Effect of GSe on the Immune Responses to aPrV Vaccine in Neonates with MatAb about 90% of Blocking Rate

Female mice (*n* = 15) were injected with 0.1 mL of InPrV vaccine twice at 14 days apart before mating with male mice. Ninety-six neonates at weaning with blocking rate of gB antibody ≥85% were selected and divided into three groups (*n* = 32/group). (a) One immunization with aP-GSe or aPrV was administered after weaning (day 21). Neonates injected with saline solution served as control. Blood samples were collected every two days from mothers and neonates during day 21 to day 69 for analysis of serum gB antibody. (b) Two immunizations with aP-GSe or aPrV were administered at 3 weeks apart. Neonates injected with saline solution served as control. Blood samples were collected every two days from mothers and neonates during day 21 to day 69 for analysis of serum gB antibody. Neonates at their 44 days old (*n* = 10/group) and 69 days old (*n* = 10 /group) were challenged with i.p. injection of fPrV (5 × 10^5^ TCID_50_).

### 2.7. Experimental Design for Effect of GSe on Cytokine Production by Neonatal Splenocytes

Neonates from immunized mothers were assigned into 3 groups (MatAb+; *n* = 6/group), and neonates from mothers injected with saline solution served as control groups (MatAb−; 3 groups, *n* = 6/group). Both MatAb+ and MatAb− groups were i.m. with aP-GSe, aPrV, and saline solution on day 21, respectively. At 12 days after the vaccination, neonate splenocytes were collected and re-stimulated by PrV antigen to detect Th1 and Th2 type cytokine productions.

### 2.8. Experimental Design for the Effect of GSe on the Primary and Secondary Immunizations

Neonates from immunized mothers were divided into five groups (*n* = 6/group). Groups 1 to 5 received 1st immunization on day 21 with aP-GSe, aP-GSe, aPrV, aPrV and saline solution, respectively, and 2nd immunization on day 42 with aP-GSe, aPrV, aP-GSe, aPrV, and saline solution, respectively (Figure 6A). Blood samples were collected from neonates before the primary immunization as well as 7, 14, 21 and 28 days after the booster immunization.

### 2.9. Assay for PrV gB Specific Antibody

Serum PrV gB specific antibody was analyzed by a commercial blocking ELSA kit (IDEXX, Inc., Westbrook, ME, USA). Briefly, 100 μL/well serum samples (diluted 1/2) were added into the antigen-coated plate. During the incubation, PrV antibodies present in the sample react with antigen on the plate. Subsequent to a wash step, 100 μL anti-PrV gB monoclonal antibody conjugate was added to each microplate well and was allowed to compete for the viral antigen. If no PrV gB antibodies are present in the test sample, the conjugate antibodies are free to react with the antigen. Conversely, if PrV gB antibodies are present in the test sample, the enzyme-conjugated monoclonal antibodies are blocked from reacting with the antigen. Following this incubation period, the unreacted conjugate was removed by washing, and 100 μL/well substrate/chromogen solution was added. Then optical density at 650 nm (OD 650) was measured using the microplate reader (Thermo Fisher Scientific, Waltham, MA, USA). Results were calculated by dividing the OD 650 of the sample by the mean OD 650 of the negative control, resulting in a sample/negative (S/N) value. The quantity of antibodies to PrV is inversely proportional to the OD 650 and thus, to the S/N value. Therefore, gB antibody level can be expressed as the blocking rate by the formula 1-S/N.

### 2.10. Experimental Design for Transcriptome Analysis

Neonates from immunized mothers were classified into 3 groups (*n* = 3/group), and then received immunization on day 21 with aP-GSe, aPrV and saline solution, respectively. After 12 days, spleen was harvested and splenocytes were isolated for transcriptome analysis by RNA-seq technology.

RNA was extracted from the splenocytes using TRIzol reagent (Thermo Fisher Scientific). RNA samples were quantified and qualified using Nanodrop 2000 Spectrophotometer (IMPLEN, Westlake Village, CA, USA) and Agilent 2100 Bioanalyzer (Agilent, Santa Clara, CA, USA), respectively. Library preparation and sequencing were performed at Novogene (Beijing, China). Illumina sequencing libraries were generated using NEBNext^®^ UltraTM RNA Library Prep Kit (Ipswich, MA, USA) following manufacturer’s recommendations [19]. The clustering of index-coded samples was based on a cBot Cluster Generation System using TruSeq PE Cluster Kit v3cBot-HS (Illumina, San Diego, CA, USA). After clustering, the library preparations were sequenced on the Illumina Hiseq platform, and then the paired-end reads with 125 bp/150 base pairs were generated.

Index of the reference genome was built using Hisat2 (v2.0.5) and paired-end clean reads were aligned to the reference genome using Hisat2 (v2.0.5). The reference genome of the program is mus musculus grcm38.p6 in the ensembl database (ftp://ftp.ensembl.org/pub/release-92/fasta/mus_musculus/). The mapped reads of each sample were assembled by StringTie (v1.3.3b) in a reference-based approach. Feature Counts (v1.5.0-p3) was used to count the reads numbers mapped to each gene. And then FPKM of each gene was calculated based on the length of the gene and reads count mapped to this gene. DESeq2 R package (1.16.1) was employed to analyze the differentially expression genes (DEGs), the resulting P values were adjusted using the Benjamini and Hochberg’s. Genes with *P* <0.05 and fold changes above 2 were defined as DEGs [19]. Additionally, Gene Ontology (GO; http://geneontology.org) and Kyoto Encyclopedia of Genes and Genomes (KEGG; http://www.genome.jp/kegg/) database were used to analyze GO and KEGG pathways of the DEGs [20,21].

### 2.11. Validation of Gene Expression by RT-qPCR

Twelve DEGs (relative fold changes were above 4Log_2_) were chosen randomly to verify the data of RNA-seq by RT-qPCR. Primers sequences used for RT-qPCR (http://www.ncbi.nlm.nih.gov/) were synthesized by Sangon Biotech (seen in Table 1). Quantitative PCR was performed using SYBR^®^Premix Ex TaqTM II (Tli RNaseH Plus) on ABI7300 (PE Applied Biosystems, Waltham, MA, USA) and determined by the 2^−ΔΔC^_T_ method [22].

### 2.12. Cytokine Assay

The assay was performed as previously described [17,23]. In brief, splenocytes were isolated from neonates under aseptic conditions and obtained in RPMI 1640 medium (Hyclone, Logan, UT, USA) supplemented with streptomycin (100 μg/mL), penicillin (100 IU/mL), and 10% fetal bovine serum (FBS; Hyclone). At a concentration of 5 × 10^6^ cells/mL, 100 μL of cells were seeded into 96-well plates and re-stimulated with inactivated PrV (5 × 10^5^ TCID_50_) antigen at 37 °C for 48 h in 5% CO_2_ atmosphere. Afterwards, the supernatants were collected and cytokines IFN-γ, IL-2, IL-4, IL-6, IL-10 and IL-12 were analyzed by ELISA kits (MultiSciences Biotech, Hangzhou, China) in accordance with the manufacturer’s instructions.

### 2.13. Statistical Analysis

GraphPad Prism 7.0 software (GraphPad Software, San Diego, CA, USA) was used to perform the data analysis. Multiple comparisons were conducted by two-way ANOVA with Tukey’s multiple comparisons test and Sidak’s multiple comparisons test or one-way ANOVA with the LSD test. Statistical differences were considered significant when the *P* value was <0.05.

## 3. Results

### 3.1. Transfer of Maternal Antibody from Mother to Neonate

To evaluate the effect of maternal vaccination on the antibody in neonates, female mice were injected with InPrV vaccine or saline solution and mated with male mice 14 days later. After birth, half of the neonates were nursed by their biological mother and the other half were nursed by non-biological mother. Figure 1B shows that serum antibody response was remarkably higher in immunized mothers than non-immunized mothers. Figure 1C shows that the amount of maternal antibody transferred from mothers to neonates mainly depended on breastfeeding. MatAb in neonates nursed by immunized mothers (P+M+; P−M+) was remarkably higher than that in neonates nursed by non-immunized mothers although their biological mother might be immunized (P+M−).

### 3.2. Effect of GSe on the Immune Responses to aPrV Vaccine in Neonates with MatAb about 50% of Blocking Rate

According to the instruction of test kit, animals with blocking rate of gB antibody ≥30% are believed positive against PrV infection. Figure 2b shows that all neonates from the day 21 to day 27 post birthday had MatAb above the protective level; MatAb declined less than the protection post day 27 in non-immunized babies; antibody level in neonates immunized with aPrV was below the protection from day 27 to day 40 (13 days); while antibody level in neonates immunized with aP-GSe was below the protection from day 27 to day 36 (9 days).

To evaluate gB antibody in protection of animals from fPrV challenge, i.p. injection with fPrV was carried out on day 28 and day 49, respectively. Figure 2c shows that all animals challenged on day 28 died within 96 h; the neonates immunized with aPrV had survival rate of 30% and those immunized with aP-GSe had survival rate of 50% after challenged on day 49. 

### 3.3. Effect of GSe on the Immune Responses to aPrV Vaccine in Neonates with MatAb about 90% of Blocking Rate

The immune responses induced by one injection of aPrV vaccine are shown in Figure 3b. Neonates non-immunized or receiving immunization of aPrV had progressively declined MatAb and the MatAb quickly dropped below the protection on days 38 and 43, respectively; neonates immunized with aP-GSe always had serum gB antibody levels above the protection throughout the experiment.

Two injections of aPrV vaccine induced significantly different antibody responses from those induced by just one injection. Figure 4b shows that neonates non-immunized or receiving immunization of aPrV had progressively declined MatAb and the MatAb quickly dropped below the protection on days 38 and 43, respectively; the antibody level rose above the protection after day 56 because of the booster immunization of aPrV with 14 days of low antibody level. However, neonates immunized with aP-GSe always had serum gB antibody levels above 30% of protection throughout the experiment.

Figure 4c shows that no animals survived in the non-immunized and aPrV groups and 30% animals survived in aP-GSe group after challenge with fPrV on day 44; no animals survived in the non-immunized groups but 30 and 60 % of animals survived in neonates immunized with aPrV and aP-GSe, respectively, after challenge on day 69.

### 3.4. Effect of GSe on Cytokine Production in Neonates with MatAb

Generally, cytokine production in MatAb+ groups was less than in MatAb−groups. Figure 5B shows that Th1 cytokines (IL-2, IL-12 and IFN-γ) were significantly less in MatAb+ group than in MatAb− group when neonates were immunized with aPrV. In addition, although Th2 cytokines (IL-4, IL_6 and IL-10) were numerically less in MatAb+ than in MatAb− groups, there was no statistical difference between two groups when aPrV was inoculated (Figure 5C). However, both Th1 and Th2 cytokine productions in MatAb+ and MatAb− groups were significantly higher in animals immunized with aP-GSe than in those immunized with aPrV. The difference for Th1 cytokine production between MatAb+ and MatAb− groups was less in aP-GSe-inoculated neonates than in aPrV-inoculated neonates. 

### 3.5. Effect of GSe on the Primary and Secondary Immunizations in Neonates

To investigate the effect of GSe on the primary and secondary immunizations, neonates were vaccinated with different combinations of aP-GSe and aPrV as shown in Figure 6A. Figure 6B shows that aP-GSe + aP-GSe and aP-GSe + aPrV groups had significantly higher antibody response than aPrV + aPrV and aPrV + aP-GSe groups after the booster immunization. Although GSe was used in the secondary immunization, it did not promote the antibody response if the primary immunization did not include GSe (aPrV + aP-GSe group). 

### 3.6. DEGs between Neonates Immunized with aP-GSe and aPrV

Figure 7 shows that there were 3976 DGEs (2357 up- and 1619 down-regulated) in aP-GSe vs. control, 5959 DGEs (2996 up- and 2963 down-regulated) in aPrV vs. control and 125 DGEs (50 up- and 75 down-regulated) in aP-GSe vs. aPrV.

### 3.7. GO Enrichment Analysis of DEGs

Analysis of GO enrichment based on DEGs is an important approach to reveal the biological process of mRNAs from three categories including the Biological Process (BP), Molecular Function (MF), and Cellular Component (CC). In the pair of aPrV vs. control (Figure 8A), terms related with immune in the catalog of BP, “antigen receptor-mediated signaling pathway”, “regulation of B cell activation”, and “regulation of leukocyte cell-cell adhesion”. For the CC, “spindle”, “condensed chromosome” and “immunoglobulin complex” were the dominant terms, and in the part of the MF section, “antigen binding” and “immunoglobulin receptor binding” were the most represented terms. As shown in Figure 8B, in aP-GSe vs. control, “activation of immune response”, “adaptive immune response”, “immune response-activating signal”, “B cell mediated immunity”, “immunoglobulin mediated immune response”, “lymphocyte mediated immunity” and “positive regulation of leukocyte activation” were the predominant terms in BP as well as “antigen binding” and “immunoglobulin receptor binding”. Furthermore, as compared to aPrV, aP-GSe could modulate the gene enrichment in immune-related terms “MHC protein complex”, “antigen binding”, “T cell receptor binding”, “calcium channel activity”, “antioxidant activity”, “antimicrobial humoral response”, “cellular lipid catabolic process”, and “antigen processing and presentation of immune cells” (Figure 8C) which indicated that GSe could modulate more immune related signaling pathways.

### 3.8. KEGG Pathway Enrichment Analysis of DEGs

The KEGG database contains the pathway maps representing the knowledge of molecular interaction and reaction networks. Compared with the control group, aPrV modulated immune related pathways including “cytokine-cytokine receptor interaction”, “T cell receptor signaling pathway”, “Th2 cell differentiation”, “C-type lectin receptor signaling pathway”, “Toll-like receptor signaling pathway”, “TNF signaling pathway”, et al. (Figure 9A). For aP-GSe vs. control, aP-GSe group could modulate more immune related pathways such as “NOD-like receptor signaling pathway”, “NF-κB signaling pathway”, “Natural killer cell mediated cytotoxicity”, “Antigen processing and presentation”, “T cell receptor signaling pathway”, “cytokine-cytokine receptor interaction”, “TGF-beta signaling pathway”, “Th1 and Th2 cell differentiation”, “p53 signaling pathway” and so on (Figure 9B).

There was different pathways enrichment in aP-GSe vs. aPrV group, such as “MAPK signaling pathway”, “Antigen processing and presentation”, “Calcium signaling pathway”, “Protein digestion and absorption” (Figure 9C), and the data of related gene was provided in Appendix A. These data indicate that GSe enhances immune responses to aPrV of neonates via involving more immune related pathways

### 3.9. Validation of DEGs by RT-qPCR

To validate the results of RNA-seq assay, a total of 12 DEGs including up-regulated or down-regulated genes were selected and measured by RT-PCR. Results from Figure 10 suggest the reliability of the RNA-seq test.

## 4. Discussion

Adjuvant effect of GSe was demonstrated on the immune response in neonatal mice. When GSe was co-administered with aPrV vaccine, specific gB antibody and cytokine (IL-2, IL-12, IFN-γ, IL-4, IL-6 and IL-10) responses were significantly increased in association with enhanced protection of vaccinated neonates against the lethal PrV challenge even though in the presence of MatAb. GSe-enhanced immune response depended on its use in the primary immunization. The mechanisms underlying may be due to more innate immune related pathways activated by GSe as analyzed by RNA-seq technology.

As the immune response of neonates and infants is poor to the infectious pathogens due to the immature immune system, MatAb acquired from the placenta and breast milk of mothers exerts an important role in protection of neonates from infection [24]. We observed that MatAb quickly declined when the neonates were nursed by mothers without immunization while MatAb sustained at a high level when the neonates were nursed by mothers immunized even though their biological mothers were not immunized (Figure 1). This result indicated that milk was the main resource for neonates to acquire MatAb. The similar results were observed by other studies. Yorty et al. reported that neonatal mice could attain herpes simplex virus specific maternal antibody from their mothers mainly by sucking breastmilk [25]. Yang et al. demonstrated that neonatal mice obtained rotavirus specific maternal antibody mainly from the milk of immunized mothers [18].

MatAb is important to protect animals from infection at early stage of life. However, MatAb declines soon in neonates after weaning, and eventually lose the protection. After reduction of MatAb and before elevation of vaccine induced antibody response, there is a window period when neonates easily attract diseases if exposed to infection [3,4,26]. We observed that all animals died after challenged with fPrV when neonatal mice were at 28 days old (Figure 2c) and at 44 days old (Figure 4c) with blocking rate of MatAb less than 30%. Injection of aPrV vaccine induced gradually increased antibody response, survival animals increased after challenged with fPrV when neonatal mice were at day 49 (Figure 2c) and day 69 (Figure 4c) with blocking rate of antibody levels more than 30%.

MatAb greatly inhibited production of antibody response induced by a vaccine. When neonates had low MatAb, aPrV-induced antibody response was already above the protective level on day 41 and then progressively increased (Figure 2b). However, aPrV-induced antibody response in neonates with high MatAb was around the protective level on day 41, and then progressively declined (Figure 4b). This difference may be due to neutralization of live viral vaccine by high level of MatAb. Crowe et al. found that viral replication of attenuated respiratory syncytial virus (RSV) vaccine was inhibited by the neutralizing effect of RSV specific MatAb in the lower respiratory tract of mice [27]. Siegrist et al. observed suppressed viral replication of live viral vaccines when MatAb was excessive, and resulted in inhibited B and T cell responses [28]. However, it is sometimes questioned to explain low immune response due to MatAb by inhibited replication of live vaccine virus because MatAb could also inhibit nonreplicating vaccine [29,30]. Johansson et al. found that suppressive effect of MatAb on rotavirus-like particle antigen was directly proportional to MatAb levels [31]. Even though the inhibitory effect of MatAb on vaccination is recognized [28,32], the exact mechanisms for MatAb to inhibit vaccination remain to be investigated. Kim et al.reported that the measles virus (MV) specific MatAb and live MV vaccine complex could cross-link the Fcγ-IIB receptor (FcγRIIB) to BCR on the surface of B cells, resulting in inhibition of B cell activation and specific antibody production in a cotton rat model [33]. Recently, Vono et al. demonstrated that MatAb could inhibit differentiation of germinal center (GC) B cells into plasma cells and memory B cells in a titer-dependent manner to interfere with vaccination in neonatal mice [34].

We previously demonstrated that GSe had adjuvant effect on aPrV vaccine [17]. Antibody response induced by aP-GSe was always higher than that induced by aPrV no matter that MatAb was low or high (Figure 2, Figure 3 and Figure 4). Surviving animals were more in neonates immunized with aP-GSe than immunized with aPrV after challenged with fPrV (Figure 2c and Figure 4c). Higher protection in aP-GSe group may be caused by higher gB antibody response [6,35], and activated NK cells and cytokine expression may also be possible contributors [36,37]. The window of susceptibility to PrV infection was shortened from 40 days to 36 days (Figure 2b) or even closed (Figure 4b) when aP-GSe was used. These results indicated that susceptibility might be greatly decreased when GSe is co-administered with aPrV vaccine.

The maternal and fetal immune system is generally believed to skew toward Th2 immune responses in order to avoid spontaneous abortion [38]. In the present study, Th1 cytokines (IL-2, IL-12, IFN-γ) but not Th2 cytokines (IL-4, IL-6, IL-10) were significantly suppressed by MatAb when neonates were inoculated with aPrV vaccine alone (Figure 5). However, both Th1 and Th2 cytokines were significantly increased when aPrV vaccine was co-administered with GSe, especially the Th1 cytokines increased by larger margin. Adjuvant enhanced Th1 response has also been found in other studies. Millan et al. observed that hepatitis B surface antigen specific Th1 immune responses were stimulated when neonatal mice were injected with antigen formulated with adjuvant CpG [39]. Honda-Okubo et al. found that neonatal mice immunized with an inactivated influenza A/H1N1 vaccine together with adjuvant AdvaxTM significantly promoted IL-2 and IFN-γ responses [40]. GSLS, the major constituent of GSe, has been found to promote Th1 cytokine production in mouse models [41,42,43]. Therefore, aP-GSe enhanced immune response found in this study could be related to increased cytokine production caused by GSLS, especially Th1 cytokine production.

Generally, the primary vaccination can affect the memory immune response. As a subtype of CD4 Th cells, T follicular helper (Tfh) cells play a critical role in GC formation and the generation of plasma cells and memory B cells [44,45,46]. Ciabattini et al. found that the improved humoral and cellular immune responses was dependent on the primary immunization after evaluation of the impact of primary and secondary immunization on the chimeric tuberculosis vaccine antigen H56 specific immune responses in mice [47]. Their previous study proved that the immune response could be elicited by a single immunization of the chimeric tuberculosis vaccine antigen H56 formulated with adjuvants, and adjuvant CAF01 and squalene-based oil-in-water emulsion could enhance Tfh and GC B cell immune responses [48]. Due to the inhibitory effect of MatAb, multiple vaccinations are usually needed in neonates to ensure a successful immunization. In this study, we observed that the antibody response was higher when the primary vaccine was aP-GSe and the antibody response was lower when aPrV was used in the primary vaccination irrespective of aP-GSe or aPrV used in the secondary immunization (Figure 6). Therefore, GSe might promote the development of Tfh and GC B cells, and generation of memory immune response. The results suggested that GSe could be included only in the primary immunization when it is used as an adjuvant to promote the immune response.

Different from the adults, the immune system of neonates is characterized by Th2 immune response bias, the low number of antigen-presenting cell (APC) and memory B cells, increased Treg cells, and reduced cytokine productions [28,49,50]. To investigate the underlying mechanisms of GSe as an adjuvant in neonates, splenocyte transcriptome was analyzed by RNA-seq technology. Data showed that the number of differentially expressed genes (DEGs) were significantly different among groups (Figure 7). The extremely high number of DEGs observed is likely due to cellular infiltration or a gross change in the composition of cell types that make up the spleen which, comprises T cells, B cells, monocytes, granulocytes, dendritic cells, natural killer cells, and macrophage, respectively (S1 and S2). Analysis of Gene ontology (GO) enrichment is a routine method to describe the gene function by identifying statistically enriched cellular components (CC), molecular functions (MF), and biological processes (BP) [21]. When compared to control, aP-GSe enriched more immune related GO terms than aPrV including “B cell activation”, “regulation of B cell activation”, “adaptive immune response”, “positive regulation of leukocyte activation”, “Lymphocyte mediated immunity” (Figure 8). Honda-Okubo et al. reported a novel adjuvant AdvaxTM could overcome the immature immunity of neonatal mice to enhance both humoral and cellular immune responses induced by A/H1N1. Moreover, it has been demonstrated that high protection provided by AdvaxTM-A/H1N1 was dependent on memory B cells [40]. Similarly, we also observed that high protection of neonates conferred by aP-GSe were associated with enhanced gB antibody responses. Hence, these data may indicate that aP-GSe could promote the development of B cells for neonates in presence of the MatAb.

Cellular immune responses are an important component of antiviral immunity. Herein, we observed that cytokine production related GO terms such as “cytokine receptor activity” as well as “positive regulation of cytokine secretion” were also enriched in aP-GSe vs. control group (Figure 8B). Some of previous animal studies reported the cellular immune responses after neonates or infants immunization in presence of MatAb were unaffected [51,52,53,54]. However, other studies showed the MatAb inhibited or reduced cellular immune responses elicited by vaccines in neonates or infants [55,56]. The discrepancy may be attributed to different kinds of vaccines or antigens. In Figure 5, aP-GSe significantly enhanced Th1 type (IL-2, IL-12, and IFN-γ) and Th2 type (IL-4, IL-6, and IL-10) cytokine productions by splenocytes of neonates with MatAb. These results demonstrated that PrV specific MatAb could inhibit the cellular immune responses of vaccinated neonates, which were similar to the previous study [56]. In addition, in comparison of aPrV, KEGG pathway “oxidative phosphorylation” was specially involved in aP-GSe (Figure 9C). Kolev et al. reported that oxidative phosphorylation played a critical role in T cell proliferation and activation [57]. Therefore, these observations indicated that GSe promoted Th1 type and Th2 type cytokine secretions even though in presence of MatAb may be attributed to the activation effect of GSe on T cells.

In association with aP-GSe promoted Th1 type and Th2 type cytokine productions, KEGG pathway analysis showed that “Th1 and Th2 cell differentiation” pathway was involved in aP-GSe vs. control (Figure 9B). BCG has been reported to elicit an effective neonatal Th1 immune responses via activating innate immunity [58,59,60], which suggested that suitable immune signals can help neonates trigger favorable immune responses. Due to defects in neonatal adaptive immune responses, activation of the neonatal innate immune responses exerts a vital role in the instruction of adaptive immunity including Th1 cytokine productions [61,62]. There are four major kinds of pattern recognition receptors expressed by APCs including toll-like receptors (TLRs), C-type lectin receptors (CLRs), Nod-like receptors (NLRs), and retinoic acid-inducible gene I (RIG-I)-like receptors (RLRs). These activated receptors could improve cytokine productions and activation of APCs to enhance the adaptive immune responses [63,64]. As a kind of APC, follicular dendritic cell (FDC) maturation is limited in neonatal mice [65,66]. However, FDC exerts an essential role in the interaction of antigen-specific T and B cells which is important for the induction of GC B and memory B cells [45]. Comparing KEGG results between aP-GSe vs. control and aPrV vs. control (Figure 9), more immune related pathways such as “NOD-like receptor signaling pathway”, “Antigen processing and presentation”, and “T cell receptor signaling pathway” were involved in aP-GSe vs. control. Particularly, we observed that KEGG pathway “antigen processing and presentation” was enriched in aP-GSe vs. aPrV. These data indicated that APCs could be activated by aP-GSe effectively. Vono et al. found that co-administered with influenza HA antigen and adjuvant CAF01 significantly enhanced GC B cell responses in neonatal mice, and which may result from the activation of CLRs [67]. Pind et al. demonstrated that adjuvants LT-K63, mmCT, MF59, and IC31 could improve the immune responses to a pneumococcal conjugate vaccine Pnc1-TT in neonatal mice via enhancing FDC maturation and GC B cell responses [68]. Therefore, aP-GSe-enhanced immune responses may result from the activation of the innate immunity of neonates.

## 5. Conclusions

In summary, neonatal mice acquired MatAb mainly from breastmilk. GSe had adjuvant effect on the immune response in neonatal mice. When GSe was co-administered with aPrV vaccine, specific gB antibody and cytokine (IL-2, IL-12, IFN-γ, IL-4, IL-6 and IL-10) responses were significantly increased in association with enhanced protection of vaccinated neonates against the lethal PrV challenge even though in the presence of MatAb. GSe-enhanced immune response depended on its use in the primary immunization. The mechanisms underlying may be due to more innate immune related pathways activated by GSe as analyzed by RNA-seq technology. These findings are valuable to use GSe as adjuvant to improve immunization using aPrV vaccine in piglets.

## Figures and Tables

**Figure 1 vaccines-08-00755-f001:**
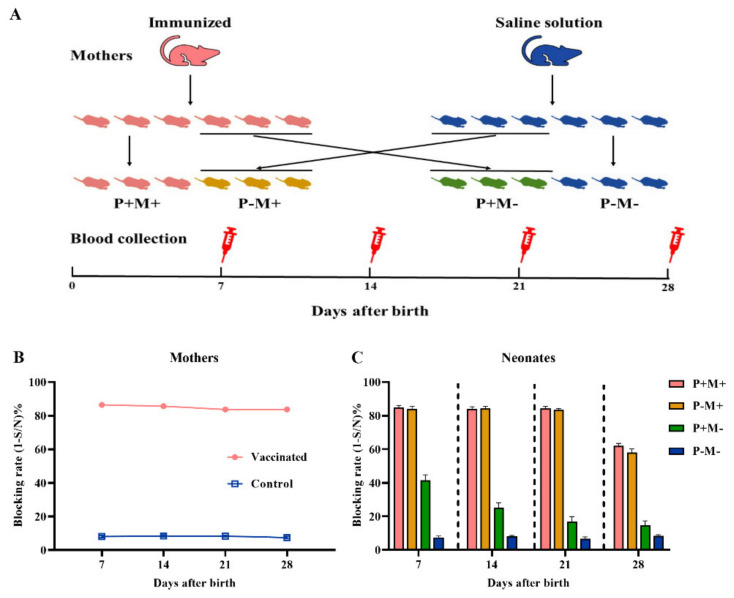
Transfer of maternal antibody from mother to neonate. (**A**) Schematic diagram for the experiment, female mice (*n* = 8) were intramuscularly (i.m.) injected with InPrV vaccine and mated with male mice (*n* = 4) 14 days later. Dams (*n* = 8) injected with saline solution in the same way served as control (non-immunized). After birth, the neonates in the same litter were divided into two groups. One group was nursed by their biological mother and the other group was nursed by non-biological mother; (**B**) Serum gB antibody responses of mothers immunized with InPrV vaccine or saline solution; (**C**) Serum gB antibody responses of neonatal mice. P+M+, nursed by their own and immunized mother; P－M+, nursed by immunized mother but their biological mother was not immunized; P+M－, nursed by non-immunized mother but their biological mother was immunized; P-M-, nursed by their own and non-immunized mother.

**Figure 2 vaccines-08-00755-f002:**
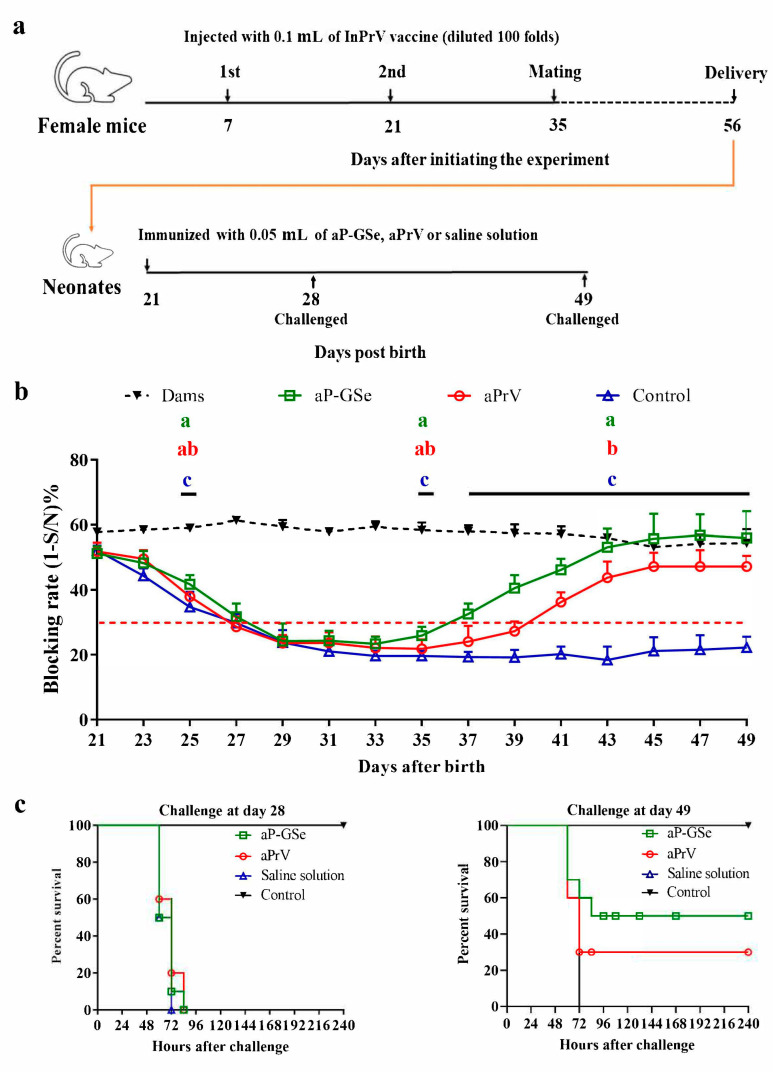
Effect of GSe on the immune responses to aPrV vaccine in neonates with MatAb about 50% of blocking rate. (**a**) Schematic diagram for the experiment; (**b**) Serum samples were collected from neonates after immunization every two days during the period of day 21 (the day of weaning) to day 49 post birth for analysis of gB antibody response by a blocking ELISA. Sample with blocking rate of gB antibody ≥30% is considered positive (red dotted line) according to the instruction of kit’s manufacturer. Data are expressed as mean ± SEM. Data with different letters in the same time point were significantly different; (**c**) The neonates (*n* = 10/group) were challenged with intraperitoneal (i.p.) injection of fPrV (5 × 10^5^ TCID_50_) on day 28 or 49 post birth. The animals were monitored for 240 h and data are presented as percent survival.

**Figure 3 vaccines-08-00755-f003:**
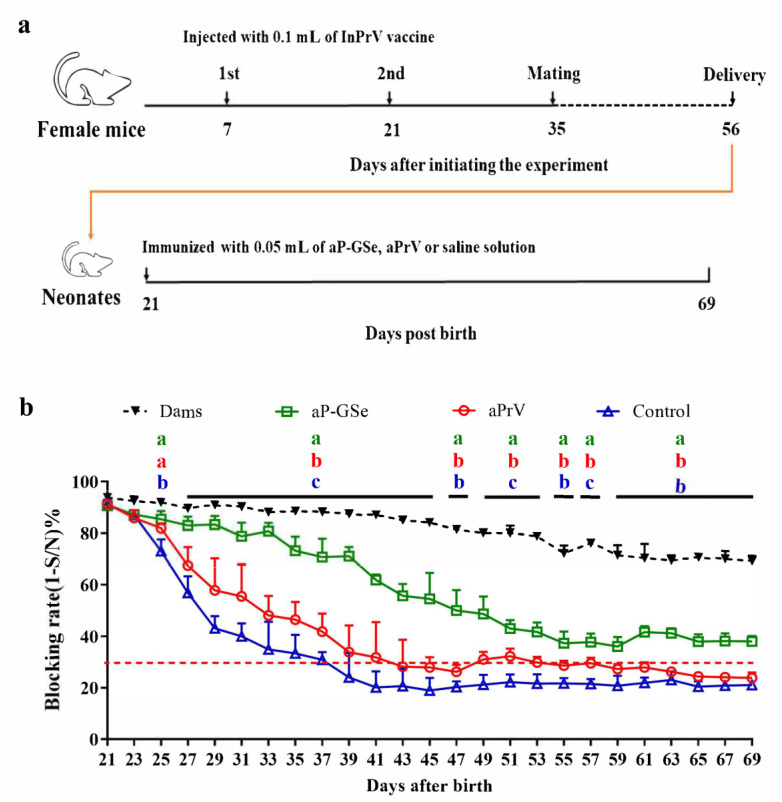
Effect of GSe on the immune response to one immunization of aPrV vaccine in neonates with about 90 % of MatAb. (**a**) Schematic diagram for the experiment; (**b**) Serum samples were collected from neonates every two days during the period of day 21 (the day of weaning) to day 69 post birth for analysis of gB antibody response by a blocking ELISA. Sample with blocking rate of gB antibody ≥30% is considered positive (red dotted line) according to the instruction of kit’s manufacturer. Data are expressed as mean ± SEM. Data with different letters in the same time point were significantly different, and the colors of letters were in line with indicated group curves (so as below).

**Figure 4 vaccines-08-00755-f004:**
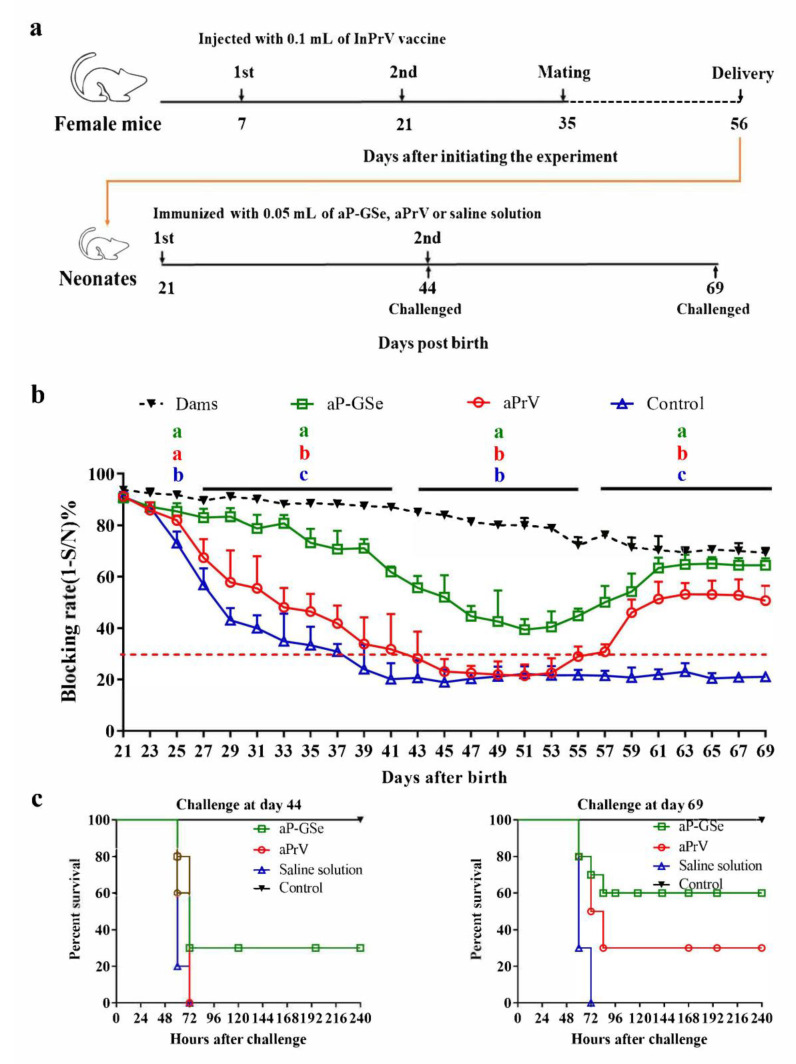
Effect of GSe on the immune response to twice immunizations of aPrV vaccine with three weeks apart in neonates with MatAb about 90% of blocking rate. (**a**) Schematic diagram for the experiment; (**b**) Serum samples were collected from neonates every two days during the period of day 21 (the day of weaning) to day 69 post birth for analysis of gB antibody response by a blocking ELISA. Sample with blocking rate of gB antibody ≥30% is considered positive (red dotted line) according to the instruction of kit’s manufacturer. Data are expressed as mean ± SEM. Data with different letters in the same time point were significantly different; (**c**) The neonates (*n* = 10/group) were challenged with intraperitoneal (i.p.) injection of fPrV (5 × 10^5^ TCID_50_) on day 44 or 69 post birthday. The animals were monitored for 240 h and data are presented as percent survival.

**Figure 5 vaccines-08-00755-f005:**
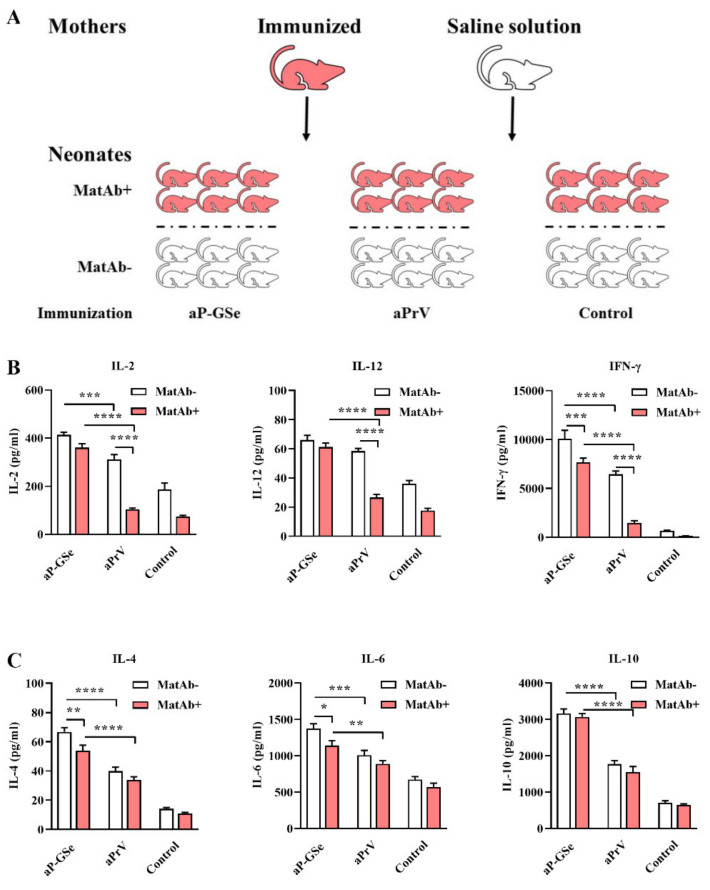
Effect of GSe on cytokine production by lymphocytes of neonates. (**A**) Schematic diagram for the experiment, neonates with MatAb (+) were from immunized female mice with InPrV vaccine and neonates without MatAb (−) were from mice injected with saline solution. Both MatAb+ and MatAb− neonates were injected with aP-GSe, aPrV or salinne solution (*n* = 6/group). The spleen was collected 12 days after immunization for isolation of splenocytes, and the cells were cultured with inactivated PrV antigen to detect (**B**) Th1 cytokines (IL-2, IL-12 and IFN-γ) and (**C**) Th2 cytokines (IL-4, IL-6 and IL-10). Data are presented as mean ± SEM. * *P* < 0.05, ** *P* < 0.01, *** *P* < 0.001, **** *P* < 0.0001.

**Figure 6 vaccines-08-00755-f006:**
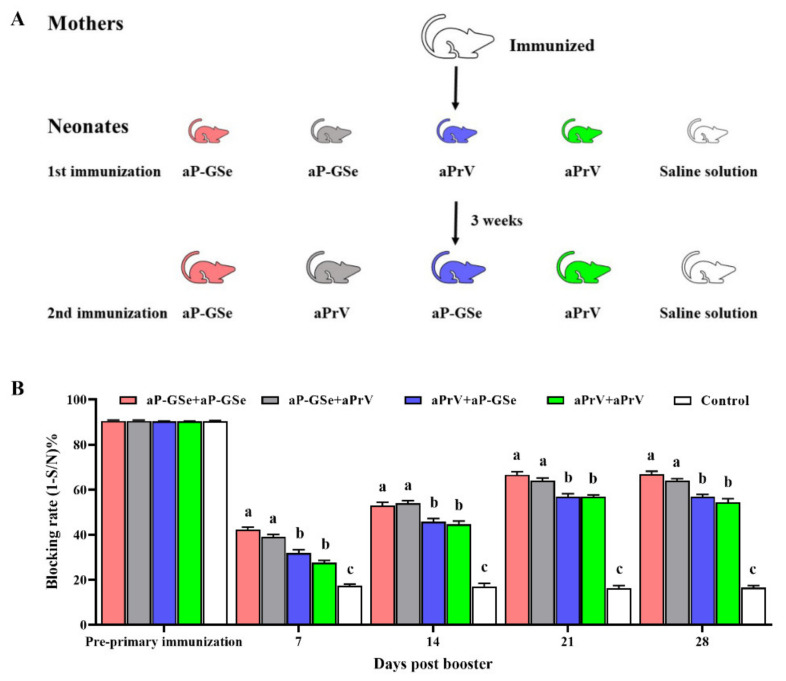
Effect of GSe on the primary and secondary immunizations in neonates. (**A**) Schematic diagram for the experiment; (**B**) Neonates with 21 days old were randomly divided into 5 groups and twice immunized at 3 weeks apart with the following vaccine combinations: groups 1, aP-GSe + aP-GSe; group 2, aP-GSe + aPrV; group 3, aPrV + aP-GSe; group 4, aPrV + aPrV; group 5, saline + saline. Blood samples were collected before the primary and after days 7, 14, 21 and 28 post the booster immunization. Data are presented as mean ± SEM. Data with significant differences among groups are shown with different letters.

**Figure 7 vaccines-08-00755-f007:**
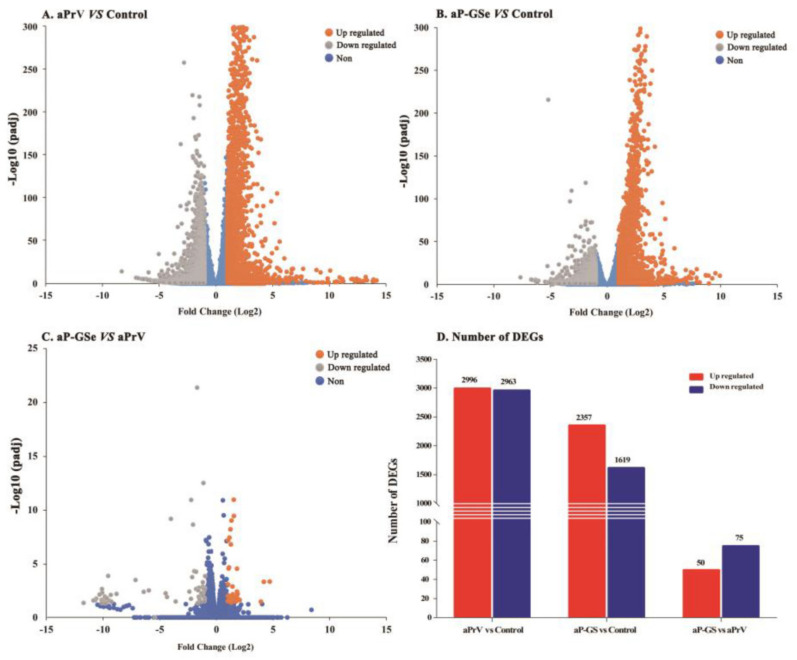
DEGs of three pairs of groups. Neonates (*n* = 3/group) at 21 days old were i.m. injected with aP-GSe, aPrV or saline solution (control). The spleen was harvested 12 days after the vaccination and splenocytes were isolated for transcriptome sequencing. Volcano plots indicate up-and down-regulated DEGs in (**A**) aP-GSe vs. control, (**B**) aPrV vs. control, and (**C**) aP-GSe vs. aPrV, (**D**) Number of DEGs.

**Figure 8 vaccines-08-00755-f008:**
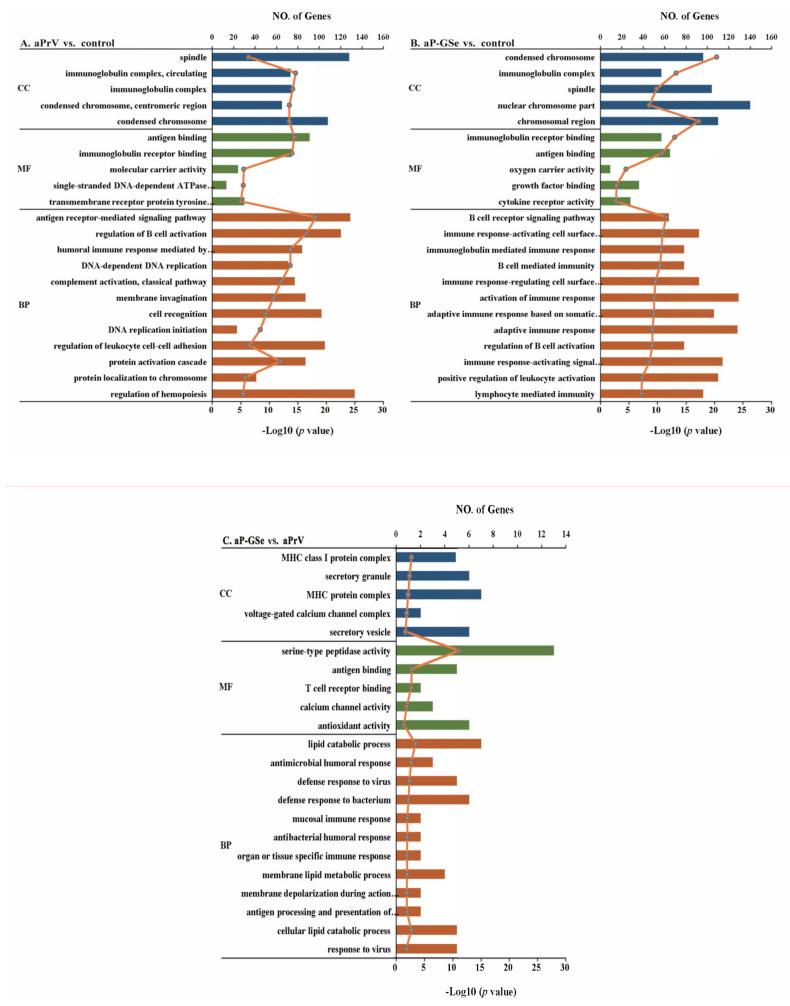
GO functional enrichment analysis. The GO terms were classified based on Gene Ontology. DEGs of (**A**) aP-GSe vs. control, (**B**) aPrV vs. control, and (**C**) aP-GSe vs. aPrV were used for GO functional enrichment analysis. GO terms with corrected *P* value less than 0.05 were considered significantly enriched by DEGs.

**Figure 9 vaccines-08-00755-f009:**
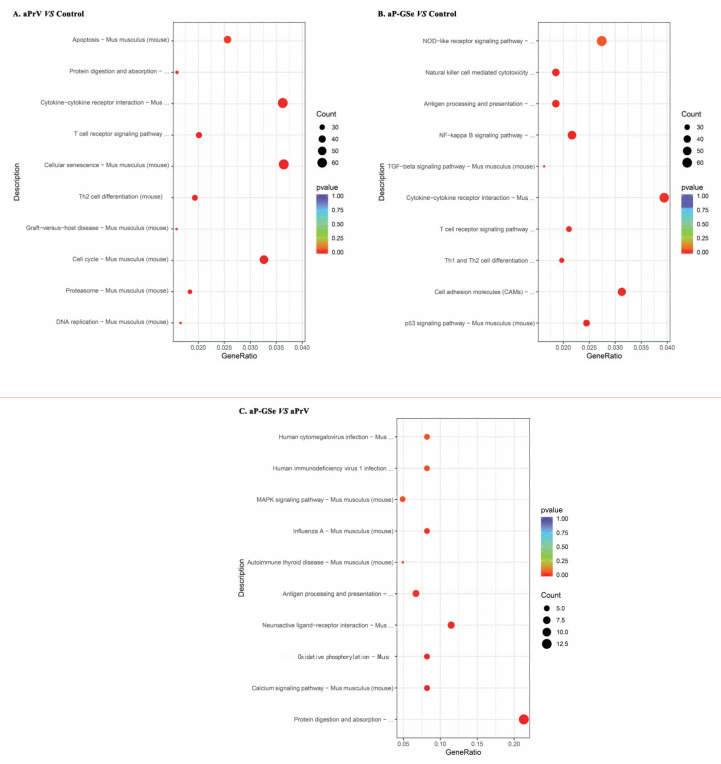
Enrichment of KEGG pathways analysis. DEGs of (**A**) aP-GSe vs. control, (**B**) aPrV vs. control, and (**C**) aP-GSe vs. aPrV were used for KEGG pathways analysis.

**Figure 10 vaccines-08-00755-f010:**
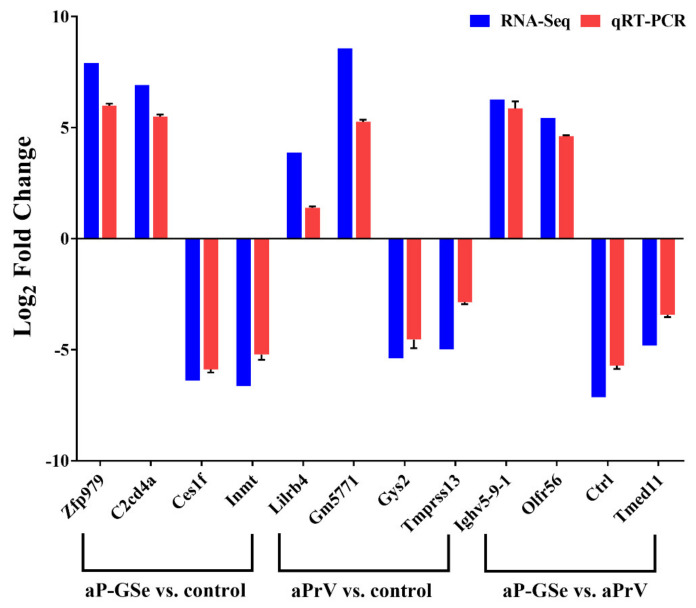
Validation of DEGs by RT－qPCR. Relative fold changes of DEGs from different comparations were normalized to the expression of GAPDH. Data are expressed as mean ± SEM.

**Table 1 vaccines-08-00755-t001:** Sequences of primers for quantitative RT-PCR.

Gene	Primer Sequence
Zfp979 (ENSMUSG00000066000)	Forward: 5-GCTGGCCTCCTAGGACATTC-3
	Reverse: 5-GGAGCAAACATTCAAGTTCTGGAT-3
C2cd4a (ENSMUSG00000047990)	Forward: 5-TCTGACTCTGAATACCAGGCAGC-3
	Reverse: 5-GGTCTGGAGTGAGCACGTTT-3
Ces1f (ENSMUSG00000031725)	Forward: 5-TGTAAGACCACCACGTCTGC-3
	Reverse: 5-TGGTCGCTATTTTTGGTATCTCCT-3
Inmt (ENSMUSG00000003477)	Forward: 5-CCTTTCTGGCCATGGAGTGT-3
	Reverse: 5-TGGAAGCGCAGAGTAACCAG-3
Lilrb4 (ENSMUSG00000112148)	Forward: 5-GGACCTGCCCTCAAGATGAC-3
	Reverse: 5-GGTTCCAGAATAAGACCTCACCA-3
Gm5771 (ENSMUSG00000058119)	Forward: 5-TCCCTGTGGATGATGATAAGATCG-3
	Reverse: 5-CACTTGGATGCGGGTTTTGT-3
Gys2 (ENSMUSG00000030244)	Forward: 5-ATCACCACCAACGACGGA-3
	Reverse: 5-GCCTCCTCTTCCTCATCATACC-3
Tmprss13 (ENSMUSG00000037129)	Forward: 5-CCAGGTCTCAGTTTCCCCAA-3
	Reverse: 5-CTCTCCAGAAGTAGAAGAGAAGG-3
Ighv5-9-1 (ENSMUSG00000095210)	Forward: 5-GAGATGGTGAATCGGCCCTT-3
	Reverse: 5-GTGCAGCCTCTGGATTCACT-3
Olfr56 (ENSMUSG00000040328)	Forward: 5-TCCTATGCTCAACCCCCTCA-3
	Reverse: 5-GGCAAACATCAGGCAACACA-3
Ctrl (ENSMUSG00000031896)	Forward: 5-ATCAGTGGTGTGGGCAATGT-3
	Reverse: 5-CATGGCATCGGTAATGCGTG-3
Tmed11 (ENSMUSG00000004821)	Forward: 5-ATCTGCTCCTGGCTTAGGAATG-3
	Reverse: 5-ATATCTAAGTGGATGCGCAGCTT-3
GAPDH (ENSMUSG00000057666)	Forward: 5-TCG TCC GGT AGA CAA AAT GG-3
	Reverse: 5-GAG GTC AAT GAA GGG GTC GT-3

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
