# Peer review of "Ginseng Stem-Leaf Saponins in Combination with Selenium Promote the Immune Response in Neonatal Mice with Maternal Antibody"

_vaccines, 2020, doi:10.3390/vaccines8040755_

Round 1

Reviewer 1 Report

This manuscript described that ginseng stem-leaf saponins in combination with selenium had adjuvant effect on the immune response to aPrV vaccine in neonates. Specific gB antibody, Th1 cytokines (IL-2, IL-12 and 17 IFN-γ) and Th2 cytokines (IL-4, IL-6 and IL-10) responses were significantly increased in association with enhanced protection of vaccinated neonates against the lethal PrV challenge. Authors described detailed experimental design and methodology. I have a few requests to the authors.

They are as follows.

  1. GenBank accession numbers for the genes listed in the Table 1 to be included.
  2. Atuthors points out that these findings are valuable to use GSe as adjuvant to improve immunization using aPrV vaccine in piglets. I would like to have a paragraph to include previous studies comparing vaccines in piglets and neonatal mice with their differences in the placental types in mind.
  3. I would like to find out whether there was any differences in fertility, or litter size when the mice were vaccinated.

In addition, I have several suggestions that are minor issues dealing with the clarification of the texts.

I describe them below with the corresponding line numbers.

Line no.          In the Text        Suggestions

25                  kyoto                       Kyoto

82                  Vetmedica, Inc.,        include city and state

86                  Rf (0.1%).                 not aligned properly.

88                  GSe                          GSe solution

134                IDEXX,                       city or state

143                Thermo Scientific,       city or state

153                Thermo fisher Scientific Fisher

155                separately                  Respectively

168                DEGs were chose        chosen

178                5 × 106 cells/ml          106

179                37º_ underline            remove underline

208                P-M-                           change order to the last

213                day of weaning                     Include day 21

240-245          No description of A, B and a, b, c     describe

Page 16          Figure 9A, Figure 9B,…                letters are not clear

401-402          difficult to follow

404                vaccination is recognized [28, 32].    may need a comma

405                Kim et al                   may need a period

408                Maria                        first name was used

413                Survival animals ---     surviving

441                enhanced ---             enhance

485                In associated with --- association

503                Vono et al ---            may need a period

532                87-+.

548                462-+.

562                9.

570                J Immunol. Res 2020, 2020.

574                Vaccines 2019, 7.

606                2015, 5.

611                1773-+.

636                MF 59 Mediates Its B    There are capital letters in the title

644                Modulation of Primary immune    There are capital letters in the title

645                2016, 7.

695                2019,2214.

Reviewer 2 Report

The study of Wanga et al. utilizes Ginseng stem-leaf saponins in combination with selenium (GSe) as an adjuvant for pseudorabies (PrV) virus vaccination. The data reveal GSe adjuvant enhances protective immune responses elicited by vaccination in neonatal mice subsequently challenged with a lethal PrV dose, even in the presence of maternal antibodies, which normally interfere with immune responses to vaccination. It was demonstrated that neonates acquire maternal antibodies from breastmilk and co-administration of GSe with aPrV vaccine enhances specific antibody and cytokine responses. GSe-enhanced immune responses were dependent on its use as an adjuvant in the primary immunization.

For animal vaccination and challenge experiments presented in figures 1-6, the experiments are correctly designed with appropriate controls, and the data presented support the conclusions drawn. For figure 7, analysis of DEGs from RNA-seq profiling, some additional information and potentially re-analysis of the data are required before acceptance. This will also probably require some additional editing of the downstream figures 8 & 9.

Major Issues:

For the RNA-seq data presented in figure 7, numbers of differentially expressed genes (DEGs) in splenocytes were calculated for 3 conditions. According to the mouse gene database, the number of mouse genes with protein sequence data = 25,059. Most cell types will express 8-12,000 genes above a cell-type specific expression threshold eg RPKM 0.5-1. The genes below this threshold represent the cellular background and comprise 1 or 2 copies per cell. In the DEG analysis presented in Fig 7, 12,064 DEGs are identified in aP-GSe vs control and 13,412 DEGs in aPrV vs controls. These numbers represent over half the protein encoded genes in the mouse genome and are clearly far too high to be correct. Even upon viral infection of a cell, which can induce significant ‘global’ transcriptional dysregulation, only 500-3,000 genes would be expected to be significantly dysregulated. To ensure that the data are correctly analysed and correctly interpreted, the following steps should be performed

  • The murine genome scaffold (eg mm9) and software used for RNA-seq mapping to the scaffold (eg Star, CLCgenomics workbench, etc) should be mentioned in the methods – currently these are missing. If the data have been mapped using outdated genome scaffolds or an inappropriate mapper (eg a DNA mapper that cannot cope with introns) then the data could be incorrectly mapped.
  • The method of expression normalization (eg RPKM) to enable comparison of different samples should be mentioned in the methods
  • It should be ensured that mapped ‘paired reads’ are only counted as a single mapping event. Otherwise mapped reads are artificially inflated.
  • After DEG analysis, FDR or Bonferoni correction for multiple comparisons should be performed. This will dramatically reduce the number of significant DEGs (<0.05)
  • Principle component analysis should be performed on each of the 3 comparisons – the controls should be clearly distinct from the aP-GSe or aPrV samples. aP-GSe vs aPrV samples should also be distinct from each other. These data should be presented.
  • A minimum expression threshold cut-off should be employed on the data (eg RPKM 0.5) to reduce false positives. For example, some genes may increase from 0 expression to 1 or 2 counts. While this is likely to be just background expression, calculation of fold change can result in huge fold changes and significant p-values (obviously FC from 0 is ∞, but the program will plot a large number here). Some filtering must be applied to remove low expression genes, where low level fluctuation is misinterpreted as dysregulation.
  • RNA-seq experiments are derived from splenocytes. Splenocytes consist of a variety of cell populations such as T and B lymphocytes, dendritic cells and macrophages, all of which have different immune functions and transcriptional profiles. Without knowledge of the cellular composition of the ‘splenocytes’, it is difficult to interpret RNA-seq data which is derived from a pool of different cell-types. Some quantification and characterization of the cell types present in the splenocytes under the different conditions – eg FACS sorting should be provided to facilitate correct interpretation of the RNA-Seq data.
  • Figure 7a-c – Labelling of the y-axes required and significance threshold should be marked.

Minor issues

  • In the introduction, it should be mentioned that pseudorabies is caused by Suid herpesvirus 1 (SuHV-1)
  • Lines 84-85: What are the abbreviations Re, Rd, etc? These should be expanded and explained.
  • Are there any chemical structures available for the active components of Ginseng stem-leaf saponins?Maybe these could be included?
  • What was the selection criteria for the test genes for qPCR? This should be mentioned.
  • In Figures 2b, 3b and 4b, it says data are expressed as mean ± SEM: however, error bars can only be seen above the mean? Lower error bars should be presented. The reporting of significant differences in these figures is also confusing and should be simplified.
  • Line 245: SEM not SE.
  • Figure 8 – in this 3 panel figure, the panels should be placed next to each other on the same page, rather than over three pages – this would allow direct comparison of the differences between conditions. The same applies to Figure 9.

Round 2

Reviewer 2 Report

I am still concerned that the RNA-seq analysis is not correctly interpreted. The extremely high number of DEGs observed is likely due to cellular infiltration or a gross change in the composition of cell types that make up the spleen which, comprises T cells (21–25%), B cells (44–58), Monocytes (3.5–5), Granulocytes (1–2), Dendritic cells (1–3%) Natural killer cells (1–2%) and Marcophages (1–2%). Without any characterization of the cell-types present, it is difficult to correctly intepret the data, which is pooled RNA from multiple cells types. The authors present no characterization of cell types present. To facilitate the correct interpretation using the RNA-seq data they have obtained, the authors should provide some expression plots (heat map or plot FPKM graphically) of selected gene markers for each of the major cell-types present in the spleen (listed above). For example, FPKM values for T-cell markers CD3, CD4, CD8 could be presented across all conditions, and compared to selected marker expression levels for the other cell types. Changes in the expression of these markers under the different conditions could then be interpreted as changes in the frequency of this cell-type. This could explaine the high numbers of DEGs recorded. This analyses should be incorporated into the manuscript before acceptance, to enable correct interpretation of the data. The data suggest there is a gross change in the cellular composition of the spleen – but currently this is not quantified.

Also, he authors also do not provide point-by-point answers to my suggestions, some are ignored or missing. Each point should be addressed. For example, the genome scaffold or mapping method have still not been added to the methods section.

In summary, the authors should present normalized expression plots of selected gene markers from major spleen cell-types, as described above, before acceptance.
